Identification of gene-phenotype connectivity associated with flavanone naringenin by functional network analysis

http://orcid.org/0000-0003-2333-3628 Fu Suhong
http://orcid.org/0000-0001-7599-6437 Zhang Yongqun yongqunzhang@yahoo.com
Shi Jing
Hao Doudou
Zhang Pengfei
Molecular Medical Laboratory, Hospital of Chengdu Office of People’s Government of Tibetan Autonomous Region , Chengdu , China
Zanin Massimiliano
Electronic publication date: 2019 Mar 19
Publication date: 2019
Volume: 7
Electronic Location ID: e6611
Received 2018 Nov 2; Accepted 2019 Feb 12
Copyright: © 2019 Fu et al.
Copyright year: 2019
Copyright holder: Fu et al.
License: This is an open access article distributed under the terms of the Creative Commons Attribution License, which permits unrestricted use, distribution, reproduction and adaptation in any medium and for any purpose provided that it is properly attributed. For attribution, the original author(s), title, publication source (PeerJ) and either DOI or URL of the article must be cited.
License URL: https://creativecommons.org/licenses/by/4.0/

Keywords: KEGG pathway, PPI network, Functional network analysis, Naringenin, Prostate cancer

Funding: The authors received no funding for this work.

==============================
Naringenin, extracted from grapefruits and citrus fruits, is a bioactive flavonoid with antioxidative, anti-inflammatory, antifibrogenic, and anticancer properties. In the past two decades, the growth of publications of naringenin in PubMed suggests that naringenin is quickly gaining interest. However, systematically regarding its biological functions connected to its direct and indirect target proteins remains difficult but necessary. Herein, we employed a set of bioinformatic platforms to integrate and dissect available published data of naringenin. Analysis based on DrugBank and the Search Tool for the Retrieval of Interacting Genes/Proteins revealed seven direct protein targets and 102 indirect protein targets. The protein–protein interaction (PPI) network of total 109 naringenin-mediated proteins was next visualized using Cytoscape. What’s more, all naringenin-mediated proteins were subject to Kyoto Encyclopedia of Genes and Genomes (KEGG) enrichment analysis by the Database for Annotation, Visualization and Integrated Discovery, which resulted in three ESR1-related signaling pathways and prostate cancer pathway. Refined analysis of PPI network and KEGG pathway identified four genes (ESR1, PIK3CA, AKT1, and MAPK1). Further genomic analysis of four genes using cBioPortal indicated that naringenin might exert biological effects via ESR1 signaling axis. In general, this work scrutinized naringenin-relevant knowledge and provided an insight into the regulation and mediation of naringenin on prostate cancer.

Introduction

Naringenin is chemically known as 5,7-dihydroxy-2-(4-hydroxyphenyl) chroman-4-one (Fig. 1) and abundantly present in grapefruits and citrus fruits and some other food items (Patel, Singh & Patel, 2018). Since naringenin was identified from extracts of dormant peach flower buds by Hendershott & Walker (1959), growing attention has been paid to naringenin-mediated bioactivities. In the past decades, a great many studies have been reported that naringenin exhibits a wide range of pharmacological activities, including beneficial effects in liver diseases (Hernández-Aquino & Muriel, 2018), antioxidative effect (Zaidun, Thent & Latiff, 2018), immunomodulating activity (Maatouk et al., 2016), and antitumor activity (Abaza et al., 2015; Gumushan Aktas & Akgun, 2018; Liao et al., 2014; Lim et al., 2017; Park et al., 2008; Sabarinathan, Mahalakshmi & Vanisree, 2011; Yen, Liu & Yeh, 2015). These findings suggest that naringenin appears to be full of therapeutic significance. Hence, it is important to explore the underlying mechanisms of its bioactivities and identify connectivity existing between gene and phenotype liaised by flavanone naringenin in publicly available data. Nevertheless, current knowledge about naringenin function and its mechanisms and targets is based on conventional experiments with high cost and long duration, which are far from integration and comprehension. Accordingly, appropriate and relevant approaches for systematic dissection of available published data are needed to generate authentic and rational lead.

Figure 1 The structure of naringenin.

Recently, various types of omics, which encompass genomics, proteomics, and other high-throughput sequencing technology, have been producing and accumulating massive data. Diverse computational approaches have been developed to mine and integrate these data in public databases, which provide researchers opportunities to conduct biomedical research, knowledge discovery, and innovative application (Lan et al., 2018). As for exploration of the relationship between naringenin-mediated proteins and their relevant phenotype, the network-based approach is considered to be a simple but effective solution, which may lie in the combination of DrugBank, the Search Tool for the Retrieval of Interacting Genes/Proteins (STRING), and the Database for Annotation, Visualization and Integrated Discovery (DAVID; Huang, Sherman & Lempicki, 2009; Szklarczyk et al., 2015; Wishart et al., 2006). DrugBank is a web-enabled tool that allows researchers to search and mine drug-primary targets information (Wishart et al., 2008). Subsequently, STRING database could predict information for primary–secondary targets interaction in silico (Szklarczyk et al., 2015).

In this study, we employed DrugBank to widely study flavanone naringenin and obtain related direct protein targets (DPTs) information. The STRING database was then used to explore the interactions between DPTs and indirect protein targets (IPTs), and protein–protein interaction (PPI) network of naringenin-mediated proteins (including DPTs and IPTs) was subsequently constructed and visualized by Cytoscape based on interaction data obtained in STRING. Next, functions of all naringenin-mediated proteins were investigated through Kyoto Encyclopedia of Genes and Genomes (KEGG) pathway enrichment analysis by DAVID. Pivotal protein genes (ESR1, PIK3CA, AKT1, MAPK1) were identified among all naringenin-mediated proteins and investigated genomic alterations using cBioPortal database. In general, the results of this functional network analysis help to provide information for integratively understanding mechanisms of naringenin action, and the implications for the prevention and therapy of prostate cancer by naringenin.

Materials and methods

Search for direct protein targets of naringenin

The DrugBank database (https://www.drugbank.ca) is a web-based bioinformatic tool, including numerous drug information such as chemical structure, targets, transporters, biointeractions, and so on (Wishart et al., 2006, 2008). The comprehensive information could be used to facilitate construction of drug-target interactome in silico. Thus, the DrugBank database was employed to search for DPTs of naringenin. The analysis results were then used for further analysis.

Network visualization and KEGG pathway enrichment analysis

Direct protein targets obtained from the DrugBank database were in turn subject to retrieval in the STRING database (http://www.string-db.org/), setting the minimum required interaction score as 0.5 and max number of interactors as 20. The IPTs of naringenin were generated after removing repetitive proteins. Then interactions of all naringenin-mediated proteins were analyzed by the STRING web server, and their PPI network was visualized using Cytoscape software (version 3.6.1) (Su et al., 2014). Subsequently, all naringenin-mediated proteins were also carried on functional KEGG pathway enrichment analysis by DAVID with p-value < 0.05.

Exploring cancer genomics data linked to naringenin by cBioPortal

The cBioPortal for Cancer Genomics (http://cbioportal.org) is an open-access platform for cancer researchers to explore multidimensional cancer genomics data (Cerami et al., 2012; Gao et al., 2013). Biologic insights and clinical applications could be understood using these rich genomic data. In this study, screened protein genes (ESR1, PIK3CA, AKT1, MAPK1) from the investigation above were subject to genetic alteration analysis in all prostate cancer studies available in cBioPortal databases. Then, the prostate cancer study with the highest genetic alteration was chosen to analyze the connectivity of the screened protein genes (ESR1, PIK3CA, AKT1, MAPK1). Gao et al.’s (2013) study helped us to interpret the result of the query in cBioPortal.

Results

Characterization of naringenin DPTs

Some small molecule chemical compounds can directly interact with proteins in organisms, which will result in alteration of the downstream pathway and various physiologic functions. In order to achieve DPTs of naringenin and relevant information, we entered Drugbank using naringenin as input. As a result, the output displayed DB03467 and total of eight primary DPTs. In the category of pharmacological activity, naringenin was classified as antiulcer agents, estrogen antagonists, BCRP/ABCG2 inhibitors, gastrointestinal agents, and hormone antagonists. Subsequent screening demonstrated seven DPTs belonging to human beings, which were presented in Table 1, ESR1, AKR1C1, CYP1B1, KANSL3, SHBG, CYP19A1, and ESR2. In addition, interactions between naringenin and seven DPTs were analyzed and illustrated in Fig. 2. It should be noted three direct target proteins (CYP19A1, SHBG, ESR2) showed a direct association with ESR1, suggesting that ESR1 played a critical role in targeted pathways controlled or mediated by naringenin.

Figure 2 The interactions of naringenin and its DPTs.

Table 1 Identification of direct protein targets of naringenin using DrugBank.

No.	Uniprot ID	Uniprot name	Gene name	
1	P03372	Estrogen receptor	ESR1	
2	Q04828	Aldo-keto reductase family 1 member C1	AKR1C1	
3	Q16678	Cytochrome P450 1B1	CYP1B1	
4	Q9P2N6	KAT8 regulatory NSL complex subunit 3	KANSL3	
5	P04278	Sex hormone-binding globulin	SHBG	
6	P11511	Aromatase	CYP19A1	
7	Q92731	Estrogen receptor beta	ESR2	

Visualization and PPI network construction of naringenin-mediated proteins

In biological systems, physiologic functions rely on interactions among diverse proteins, which can be represented by a network consisting of nodes and edges. Thus, there should be of great importance to search for downstream targets of naringenin DPTs. To this end, STRING database was applied to identify DPTs-related proteins and the results were summarized in Table S1. In general, total of 109 naringenin-mediated proteins (Table S2) were generated including seven DPTs and 102 IPTs. Moreover, the interaction data of all naringenin-mediated proteins were obtained using STRING (Table S3) and the PPI network were then visualized by Cytoscape 3.6.1 (Su et al., 2014). As shown in Fig. 3, there were 912 PPI pairs in the network of naringenin-mediated proteins. The nodes indicated proteins and the edges indicated the interactions between these proteins. Next, node degree was evaluated to investigate the centrality of proteins and represented by size, low values to the small size. Ranked by degree value from large to small, top 10 proteins were screened out and showed in Table 2, including three DPTs (ESR1, CYP19A1, ESR2). These findings collectively indicated that biological effects mediated by naringenin are connected to ESR1.

Figure 3 PPI network of naringenin-mediated proteins.

The nodes indicate proteins and the edges indicate interaction between proteins. High node degree value was represented by big size and low node degree value was represented by small size.

Table 2 The list of top 10 proteins ranked by degree value.

No.	Gene name	Node degree	No.	Gene name	Node degree	
1	ESR1	51	6	HPGDS	38	
2	CYP19A1	48	7	ESR2	35	
3	HSD17B6	44	8	JUN	34	
4	CYP1A1	43	9	HSD3B2	34	
5	CYP3A4	38	10	UGT1A6	33	

KEGG enrichment pathway analysis of naringenin-mediated proteins

As a collection of molecular interaction, reaction, and relation networks, KEGG pathways offer a good understanding of high-level functions and the biological system. Thus, we performed the KEGG pathway enrichment analysis using DAVID to assess the functional feature of naringenin-mediated proteins (Table S4). As can be seen in Table 3, the top 15 KEGG pathways linked to naringenin-mediated proteins were identified and included steroid hormone biosynthesis, metabolism of xenobiotics by cytochrome P450, chemical carcinogenesis, drug metabolism-cytochrome P450, retinol metabolism, pentose and glucuronate interconversions, ascorbate and aldarate metabolism, ovarian steroidogenesis, drug metabolism-other enzymes, porphyrin and chlorophyl metabolism, prolactin signaling pathway, thyroid hormone signaling pathway, estrogen signaling pathway, metabolic pathways, and prostate cancer. The broad grouping of functional enrichment analysis results indicated that naringenin-mediated proteins were mainly linked to (1) estrogen-related signaling pathways, (2) basal metabolism pathways, (3) cancer-related pathways. Given the importance of ESR1 in all naringenin-mediated proteins, emphasis of biological functions was directed to pathway possessing ESR1. Thus, three ESR1-related pathways were identified: prolactin signaling pathway, thyroid hormone signaling pathway, and estrogen signaling pathway. Besides, because anticarcinogenic properties of naringenin have been reported in diverse malignant tumors as mentioned above, prostate cancer pathway consisting of eight proteins was found: AKT1, MAPK1, IGF1R, AR, CCND1, INS, PIK3CA, IGF1. This result suggested that prostate cancer might be as a phenotype connected to naringenin-mediated proteins.

Table 3 Top 15 enriched KEGG pathways identified using DAVID.

Pathway description	Gene count	Gene	p-value	
Steroid hormone biosynthesis	33	CYP3A4, HSD3B2, CYP3A5, HSD3B1, CYP1B1, HSD17B2, HSD17B1, CYP11B1, COMT, UGT1A7, AKR1C3, UGT1A6, UGT1A9, UGT1A3, UGT1A4, UGT2A1, HSD17B6, HSD17B3, SRD5A1, UGT2A3, SULT1E1, HSD17B7, AKR1C1, CYP19A1, HSD17B8, CYP1A1, UGT1A1, UGT1A10, UGT2B17, CYP17A1, UGT2B15, AKR1D1, UGT2B7	4.86E-49	
Metabolism of xenobiotics by cytochrome P450	26	GSTA1, CYP3A4, CYP3A5, CYP1B1, SULT2A1, CYP1A1, EPHX1, GSTT1, UGT1A1, DHDH, GSTM1, UGT1A7, UGT1A10, GSTM2, UGT1A6, UGT1A9, UGT2B17, GSTM3, UGT1A3, UGT1A4, UGT2A1, UGT2A3, UGT2B15, UGT2B7, AKR1C1, GSTP1	3.29E-31	
Chemical carcinogenesis	24	GSTA1, CYP3A4, CYP3A5, CYP1B1, SULT2A1, CYP1A1, EPHX1, GSTT1, UGT1A1, UGT1A7, GSTM1, UGT1A10, UGT1A6, GSTM2, UGT1A9, UGT2B17, GSTM3, UGT1A3, UGT1A4, UGT2A1, UGT2A3, UGT2B15, UGT2B7, GSTP1	8.08E-27	
Drug metabolism-cytochrome P450	20	GSTA1, CYP3A4, CYP3A5, GSTT1, UGT1A1, UGT1A7, GSTM1, UGT1A10, UGT1A6, GSTM2, UGT1A9, UGT2B17, GSTM3, UGT1A3, UGT1A4, UGT2A1, UGT2A3, UGT2B15, GSTP1, UGT2B7	6.4E-22	
Retinol metabolism	16	CYP3A4, CYP3A5, CYP1A1, UGT1A1, UGT1A7, UGT1A6, UGT1A10, UGT1A9, UGT2B17, UGT1A3, UGT1A4, UGT2A1, HSD17B6, UGT2A3, UGT2B15, UGT2B7	3.57E-16	
Pentose and glucuronate interconversions	13	UGT1A7, UGT1A10, UGT1A6, UGT1A9, UGT2B17, UGT1A3, UGT1A4, UGT2A1, UGT2A3, UGT2B15, UGT1A1, DHDH, UGT2B7	1.02E-15	
Ascorbate and aldarate metabolism	12	UGT1A7, UGT1A10, UGT1A6, UGT1A9, UGT2B17, UGT1A3, UGT1A4, UGT2A1, UGT2A3, UGT2B15, UGT1A1, UGT2B7	3.81E-15	
Ovarian steroidogenesis	14	AKR1C3, HSD3B2, IGF1R, CYP17A1, HSD3B1, CYP1B1, CYP1A1, HSD17B2, INS, HSD17B1, IGF1, GNAS, HSD17B7, CYP19A1	6.86E-15	
Drug metabolism-other enzymes	13	CYP3A4, UGT1A7, UGT1A10, UGT1A6, UGT1A9, UGT2B17, UGT1A3, UGT1A4, UGT2A1, UGT2A3, UGT2B15, UGT1A1, UGT2B7	1.01E-13	
Porphyrin and chlorophyl metabolism	12	UGT1A7, UGT1A10, UGT1A6, UGT1A9, UGT2B17, UGT1A3, UGT1A4, UGT2A1, UGT2A3, UGT2B15, UGT1A1, UGT2B7	1.10E-12	
Prolactin signaling pathway	12	AKT1, MAPK1, FOS, CYP17A1, CCND1, INS, ESR1, PIK3CA, MAPK11, ESR2, PRL, SRC	4.99E-10	
Thyroid hormone signaling pathway	12	AKT1, MAPK1, NCOA1, CCND1, NCOA2, NCOA3, ESR1, PIK3CA, NCOR1, MYC, SRC, MED1	9.09E-08	
Estrogen signaling pathway	10	AKT1, MAPK1, FOS, JUN, ESR1, PIK3CA, NOS3, GNAS, ESR2, SRC	2.26E-06	
Metabolic pathways	32	CYP3A4, HSD3B2, CYP3A5, HSD3B1, HSD17B2, HSD17B1, CYP11B1, COMT, AKR1C3, UGT1A7, UGT1A6, UGT1A9, POLE4, UGT1A3, UGT1A4, UGT2A1, HSD17B6, HSD17B3, NOS3, UGT2A3, HPGDS, HSD17B7, CYP19A1, HSD17B8, CYP1A1, UGT1A1, UGT1A10, UGT2B17, CYP17A1, UGT2B15, AKR1D1, UGT2B7	1.18E-05	
Prostate cancer	8	AKT1, MAPK1, IGF1R, AR, CCND1, INS, PIK3CA, IGF1	7.70E-05	

Genetic alterations connected with naringenin-mediated protein genes in prostate cancer

Previous functional enrichment analysis uncovered the link between naringenin-mediated proteins and prostate cancer pathway. Further exploration was needed to validate this link. Since ESR1 enjoyed the highest centrality among all naringenin-mediated proteins as proved previously, and because three overlapping proteins (PIK3CA, AKT1, MAPK1) associated with ESR1 were found to be connected to prostate cancer pathway as well, the genomic alteration of these four protein genes (ESR1, PIK3CA, AKT1, MAPK1) were checked in prostate cancer using cBioPortal. Among 16 prostate cancer studies (Baca et al., 2013; Barbieri et al., 2012; Beltran et al., 2016; Fraser et al., 2017; Gao et al., 2014; Grasso et al., 2012; Hieronymus et al., 2014; Knudsen & Scher, 2009; Cancer Genome Atlas Research Network, 2015; Robinson et al., 2015; Taylor et al., 2010), gene alterations ranged from 0% to 31.58% as displayed in Fig. 4. Because of the most pronounced genetic alterations among all prostate cancer studies available in cBioPortal, the NEPC study (Beltran et al., 2016) was selected individually to view relevant genomic changes of four genes. A concise and compact summary of alterations in three queried genes (ESR1, PIK3CA, AKT1, MAPK1) was shown in Fig. 5 using OncoPrint. The results presented that most gene alterations belonged to amplification. Additional mutual exclusivity analysis indicated that every gene pair exhibited significant (p-value < 0.05) co-occurrence in prostate samples in the study of NEPC (Table 4). The co-occurrence of ESR1 and other three genes revealed a central axis function for ESR1 under naringenin control.

Figure 4 Overview of changes on ESR1, PIK3CA, AKT1, and MAPK1 genes in genomics data sets available in 16 different prostate cancer studies.

Figure 5 A visual summary of alteration across a set of prostate samples (data taken from the NEPC studies, Nat Med 2016) (Beltran et al., 2016) based on a query of four genes ESR1, PIK3CA, AKT1, and MAPK1.

Different genomic alterations are summarized and color coded presented by % changes in particular affected genes in individual tumor samples. Each row represents a gene, and each column represents a tumor sample.

Table 4 Mutual exclusivity analysis of four selected genes (ESR1, AKT1, PIK3CA, MAPK1) in NEPC study.

Gene A	Gene B	p-value	Log2 odds ratio	Association	
ESR1	AKT1	<0.001	>3	Tendency toward co-occurrence	
ESR1	PIK3CA	<0.001	>3	Tendency toward co-occurrence	
ESR1	MAPK1	0.003	>3	Tendency toward co-occurrence	
PIK3CA	AKT1	<0.001	2.733	Tendency toward co-occurrence	
AKT1	MAPK1	0.002	>3	Tendency toward co-occurrence	
PIK3CA	MAPK1	0.005	>3	Tendency toward co-occurrence	

Subsequently, the Network tab embedded in cBioportal was exploited to explore the interactive relationship between four selected genes and genes that were altered in NEPC prostate cancer study. A query of ESR1, PIK3CA, AKT1, and MAPK1 automatically generated a network containing 50 neighbor genes of four query genes, and legends were available to explain network symbols (Fig. 6A). To manage network complexity, we filtered neighbors by 45% alteration, such that only AR gene with the highest alteration frequency remained in addition to four query genes (Fig. 6B). The pruned network revealed the potential interactions between naringenin-mediated genes and altered genes in prostate cancer samples. Moreover, specific cancer drugs acting on ESR1, PIK3CA, AKT1, and MAPK1 were displayed in Fig. 5. ESR1 was the main target of most FDA approved drugs (represented by yellow hexagon) in the network, providing a molecular basis for potential clinical applications of naringenin to treat prostate cancer targeting ESR1.

Figure 6 A visual display of the gene network connected to ESR1/PIK3CA/AKT1/MAPK1 in prostate adenocarcinoma (based on the NEPC study, Nat Med 2016) (Beltran et al., 2016).

(A) Cross-cancer alteration summary for ESR1/PIK3CA/AKT1/MAPK1 mined from the cBioPortal for Cancer Genomics. Multidimensional genomic details are shown for seed genes ESR1, PIK3CA, AKT1, and MAPK1. A darker shade of red indicates increased frequency of alteration in prostate cancer. (B) Neighboring genes connected to the four query genes, filtered by alterations (45%).

Discussion

Since naringenin was reported by Hendershott & Walker (1959), there has been an increasing number of publications on naringenin and more than 2,400 publications have accumulated on PubMed to date. A wide range of biological and cellular activities have been reported for naringenin, including relevant targets and biological pathways. These findings suggested that naringenin connected with a plethora of diseases and possessed a fascinating nature. However, there are still barriers to systematically understand how naringenin facilitates its wide range of beneficial effects. As such, a bridge between naringenin and its direct and indirect targets needs to be established. To this end, functional network analysis (Hsieh et al., 2016; Shi et al., 2017) with new analytical ways or platforms is introduced in this study and enables to explore naringenin-mediated proteins, thereby elucidating connectivity between protein genes and phenotype.

In our study, functional network analysis was performed by using a series of web-based bioinformatic tools. Firstly, the feasibility of analysis for connectivity between naringenin targets and its phenotypes was demonstrated by using DrugBank and STRING, resulting in the identification of seven DPTs (ESR1, AKR1C1, CYP1B1, KANSL3, SHBG, CYP19A1, ESR2), 102 IPTs (Table S2). Secondly, the KEGG enrichment analysis conducted by DAVID identified three ESR1-related signaling pathways and prostate cancer pathway, which were significantly altered by naringenin-mediated proteins. As supporting evidence, previous studies have observed that naringenin exhibits antineoplastic property against most solid tumors including breast and colorectal (Abaza et al., 2015), bladder (Liao et al., 2014), prostate (Lim et al., 2017), and so on. Thus, the association between prostate cancer and the beneficial effects exerted by naringenin in cancer was then explored and assessed by the genetic alternations in four protein genes (ESR1, PIK3CA, AKT1, and MAPK1) which were revealed by naringenin associated three ESR1-related signaling pathways and prostate cancer pathway. Most of the genetic alterations in ESR1, PIK3CA, AKT1, and MAPK1 were amplifications, suggesting an excess expression in prostate cancer. Consistent with our study, Fu et al. (2014) pointed out it that the genetic polymorphisms in ESR1 gene could cause transcription change, resulting in the influence the risk of prostate cancer. PI3K/AKT pathway including PIK3CA and AKT1 is a well-known pathway involved in the regulation of cell proliferation, metastasis, and apoptosis (Mayer & Arteaga, 2016). Recent research even found that PIK3CA amplification had a correlation with poor survival of patients with prostatic carcinoma (Pearson et al., 2018). To the present, MAPK1 is a famous oncogene, acting as a signal transduction node of various upstream signals such as proliferation, inhibition of apoptosis, and so on (Chen et al., 2015). Moreover, the mutual exclusivity analysis discovered a tendency toward co-occurrence between ESR1 and three overlapping genes. Hence, the results were in good agreement with it that ESR1 acted as a major driver of anti-carcinogenic efficiency in prostate cancer. In addition, PIK3CA, AKT1, and MAPK1 directly interacted with AR (Fig. 6B), which was a naringenin-mediated protein gene as well and participated in prostate cancer pathway. As support, AR has been reported to be crucial to prostate cell growth and development (Heinlein & Chang, 2002). Therefore, a hypothesis is currently proposed that naringenin acts through a series of genes (ESR1, PIK3CA, AKT1, MAPK1, and AR) to control prostate cancer cell proliferation.

Prostate cancer is the most common malignancy in male cancer patients with high mortality, accounting for 23% of all new cancer cases worldwide (Siegel, Miller & Jemal, 2018). The gene-phenotype connectivity liaised by naringenin revealed that the regulation of ESR1 by naringenin might be a major driver for the chemoprotective salutary effects on prostate adenocarcinoma. The candidate genes identified may facilitate the comprehension of genomic results and be considered to provide information useful for guiding further research of naringenin and future drug development. However, some challenges of this analysis remain to be investigated and solved: (1) whether the connectivity existing between naringenin and prostate cancer can extend to other malignant tumors is still essential to explore; (2) the role of genes under naringenin control detected in this research must be verified in prostate cancer.

Conclusions

In conclusion, functional network analysis based on a series of online tools including DrugBank, STRING, DAVID, and cBioPortal has been applied to mine and integrate knowledge of naringenin biological action. The retrieval of publicly available computational databases unraveled the connectivity existing between naringenin and prostate cancer. A hypothesis currently being considered is that naringenin could regulate prostate cancer via ESR1 signaling axis. Naringenin might be promising as an alternative chemotherapy or chemoprevention for prostate cancer. Overall, this functional network analysis provided a reasonable hypothesis and assisted in the acceleration of naringenin biology research.

Supplemental Information

Supplemental Information 1 Direct protein targets and indirect protein targets of naringenin.

Click here for additional data file.

Supplemental Information 2 List of 109 naringenin-mediated proteins.

Click here for additional data file.

Supplemental Information 3 Interactions of 109 naringenin-mediated proteins.

Click here for additional data file.

Supplemental Information 4 All Enrichment KEGG pathway.

Click here for additional data file.

Additional Information and Declarations

Competing Interests

Author Contributions

Data Availability

The authors declare that they have no competing interests.

Suhong Fu conceived and designed the experiments, performed the experiments, analyzed the data, contributed reagents/materials/analysis tools, prepared figures and/or tables, authored or reviewed drafts of the paper, approved the final draft.

Yongqun Zhang conceived and designed the experiments, analyzed the data, authored or reviewed drafts of the paper, approved the final draft.

Jing Shi analyzed the data, prepared figures and/or tables, authored or reviewed drafts of the paper, approved the final draft.

Doudou Hao prepared figures and/or tables, authored or reviewed drafts of the paper, approved the final draft.

Pengfei Zhang prepared figures and/or tables, authored or reviewed drafts of the paper, approved the final draft.

The following information was supplied regarding data availability:

The raw measurements are available in the Supplemental Files.

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
