# Peer review of "Identification of gene-phenotype connectivity associated with flavanone naringenin by functional network analysis"

_PeerJ, doi:10.7717/peerj.6611_

## Round 0.1 · original submission · Major Revisions

Dear authors,

Please find below the reports of two Referees. I would advise to put special care in the description and explanation of the methodology; especially in the justification of some methodological steps, like the threshold of 25 in the degree.

Reviewer 1 ·

Basic reporting

1.The English language should be improved and the manuscript needs professional care. For example, in Line 21 “consequences showed that naringenin...” should be “the results showed that...”
2. The legends for the figures should be better described. In figure 4, what’s the difference between the links with and without arrows?

Experimental design

1.The PPI network construction of naringenin-mediated proteins is not sufficiently described. The STRING web server has provided the PPI network for the input protein list. But in line 77-78 and 109, the PPI network was constructed by Cytoscape. Did the authors used any special plugin or modules of Cytoscape to construct the PPI network? Or just visualize the network? Please make it more clear in methods part.
2.Why did the authors select the hub proteins by node degree larger than 25? It’s a relative high degree, please give more explanation.

Validity of the findings

Why just prostate cancer was chosen as the phenotype associated with naringenin-mediated proteins? Is it just because of hormone related? If so, why other types of cancer that hormone related were not considered, such as breast cancer.

Annotated reviews are not available for download in order to protect the identity of reviewers who chose to remain anonymous.

Reviewer 2 ·

Basic reporting

no comment

Experimental design

no comment

Validity of the findings

Speculation is welcome, but should be identified as such.

Additional comments

This paper used a set of bioinformatics tools to integrate available data on naringenin, and find some prostate cancer related genes. The authors proposed that they find gene-phenotype connectivity. What is the genetype and phenotype here? PPI network and prostate cancer? I don’t think the relationship between PPI and prostate cancer is the most important finding here. So, I suggest using a more specified title. More comments are listed in the following:
1. The first step is to put naringenin into the 8 targets returned by DrugBank without screening. Among them, the ttgR gene belongs to Pseudomonas putida species and the other seven genes belong to human. I was wondering if is more reasonable to not include ttgR gene in the further analysis, as prostate cancer is the human disease
2. In the network analysis, degree > 25 was used to select key proteins. This analysis seems too simple, and the authors may use more network parameters to make their finding more convincing.
3. What the network scale used for pathway analysis? STRING can give networks with different scales. Please clarify it
4. There are various English problems. It it better to have a proofreading service.

---

## Round 0.2 · accepted · Accept

While the paper has substantially improved, I still agree with the first Referee, in that there are some strange sentences throughout the manuscript. I would therefore suggest some language editing before its publication.

# # Reviewer 1 ·

Basic reporting

no comment

Experimental design

no comment

Validity of the findings

no comment

Additional comments

The authors have addressed all my comments and have accordingly improved the presentation of their manuscript. But there are still some English problems, please make improvement.

Reviewer 2 ·

Basic reporting

has been improved

Experimental design

has been improved

Validity of the findings

has been improved